# Ultrafast 3D printing with submicrometer features using electrostatic jet deflection

Ievgenii Liashenko [1,2], Joan Rosell-Llompart [1,3]* & Andreu Cabot [2,3]*

Additive manufacturing technologies based on layer-by-layer deposition of material ejected from a nozzle provide unmatched versatility but are limited in terms of printing speed and resolution. Electrohydrodynamic jetting uniquely allows generating submicrometer jets that can reach speeds above $1\,\mathrm{m\,s^{-1}}$, but such jets cannot be precisely collected by too slow mechanical stages. Here, we demonstrate that controlling the voltage applied to electrodes located around the jet, its trajectory can be continuously adjusted with lateral accelerations up to $10^6\,\mathrm{m\,s^{-2}}$. Through electrostatically deflecting the jet, 3D objects with submicrometer features can be printed by stacking nanofibers on top of each other at layer-by-layer frequencies as high as 2000 Hz. The fast jet speed and large layer-by-layer frequencies achieved translate into printing speeds up to $0.5\,\mathrm{m\,s^{-1}}$ in-plane and $0.4\,\mathrm{mm\,s^{-1}}$ in the vertical direction, three to four orders of magnitude faster than techniques providing equivalent feature sizes.

[1] Department of Chemical Engineering, Universitat Rovira i Virgili, Av. dels Països Catalans 26, 43007 Tarragona, Spain. [2] Catalonia Institute for Energy Research - IREC, Jardins de les dones de negre 1, Sant Adrià de Besòs, Barcelona 08930, Spain. [3] Catalan Institution for Research and Advanced Studies - ICREA, Pg. Lluís Companys 23, 08010 Barcelona, Spain. *email: joan.rosell@urv.cat; acabot@irec.cat

Additive manufacturing has become the new paradigm of distributed production of customized products, providing advantages in terms of geometric freedom of design, material utilization, and lead time reduction[1–3]. However, existing additive manufacturing technologies still have important limitations, especially on production speed, availability and combination of materials, and control over their microstructure and thus functionality. Additionally, the cost and complexity of manufacturing equipment that enables producing submicrometer features are prohibitive for a true distributed production.

In additive manufacturing strategies based on ejection of a melt or an ink from a nozzle, material is selectively deposited layer-by-layer either as a continuous extruded filament or jet, or as a train of droplets formed by the actuation of a piezoelectric, a thermal, an ultrasonic or an electrostatic mechanism[4,5]. Besides cost and simplicity, the main advantage of nozzle-based 3D printing strategies is that they allow the manufacture of items made of virtually any substance, ranging from polymers[6,7], to metals[8,9], to ceramics[10], to wood[11,12], and even to biological tissues[13]. Such unmatched material versatility stems from the use of metal or polymer melts or solvent-based inks, which can be formulated to contain any component in the form of ions, molecules, nanoparticles, or even living cells[14–17].

However, current nozzle-based 3D printing technologies are relatively slow as they rely on the use of a mechanical stage to define the geometry of the printed product. Another drawback is a limited printing resolution because the width of the printed lines correlates with that of the nozzle aperture, typically well above several tens of micrometers. Nozzles with smaller apertures suffer from frequent clogging and high viscous losses, and, consequently, can only be used with low-viscosity inks free of large particles, which obviously limits material versatility. To make matters worse, the smaller the jet thickness gets, the slower the printing process.

To effectively reduce the width of the printed lines, instead of forcing the extrusion of material through very thin nozzles, the melt or ink can be pulled by means of an electric field applied between the nozzle and the printing substrate. Once the electrical stresses acting on the liquid surface overcome the surface tension stress, the liquid meniscus forms a Taylor cone wherefrom a thin and fast jet of ink is propelled electrostatically towards the printing substrate[18] (Fig. 1a). Such electrohydrodynamic (EHD) jetting strategy is uniquely suited for high resolution 3D printing compared to other nozzle-based 3D printing methods[19–24]. EHD jetting allows printing submicrometer features with no risk of nozzle clogging, as it enables the generation of nanometer-sized jets from wide nozzle apertures using a great variety of inks, with viscosities ranging over several orders of magnitude[25].

Despite its advantages, EHD jetting has not been developed to its full potential for 3D printing because of the huge challenge that the precise location of very fast electrified jets, with speeds that can be in excess of $1 \, \text{m} \, \text{s}^{-1}$, presents[26]. Current systems based on EHD jetting use mechanical stages to locate the material on the printing substrate. However, mechanical stages can only match the huge speeds of the electrified jets in long straight lines, but cannot attain the giant accelerations that are needed to sustain such speeds while printing small complex patterns. These unmatched speeds result in uncontrolled jet buckling onto the substrate, degrading the quality of the printed pattern. Whereas buckling has been used to print simple cycloid and wavy patterns along straight lines[21], the minimum radius of curvature that can be printed without buckling is limited to around 0.5 mm when using high speed electrified jets ($>0.1 \, \text{m} \, \text{s}^{-1}$).

To bypass the challenge of precisely collecting such fast jets, the overall material flow rate has been reduced by pulsing the jet generation[27]. This strategy, known as EHD drop-on-demand printing, has allowed printing resolutions down to 50–80 nm[28]. However, the severe decrease in printing speed due to the pulsing may make this strategy too slow for industrial implementation.

To unleash the potential of high-speed printing allowed by very fast EHD jetting, here we propose to electrostatically deflect the jet trajectory as a suitable strategy to control the location of a continuous electrified jet when it impacts the substrate. Electrostatic deflection of electrified jets have been previously attempted in simple configurations: (i) patterning the substrate to direct the jet toward specific locations[29–31]; (ii) manual switching of DC potentials[32,33]; and (iii) using elementary wavefunctions[34–41]. But all of them lack the fast control necessary for 3D printing of complex patterns. Here we demonstrate that through high speed amplifiers controlled by software, ultrafast jet deflection can be achieved and used to print 3D objects with submicrometer features and very small radius of curvature.

## Results

**3D printer design.** A conventional EHD printer basically comprises a nozzle, a system which supplies the printable material to this nozzle such as a syringe pump, a printing substrate mounted on a XY mechanical stage, and a high voltage power supply connected in between the nozzle and the substrate. The printing substrate is usually placed onto an electrically-grounded electrode and it is preferentially conductive or it has a conductive surface to dissipate electrostatic charge fast enough. The required charge dissipation time depends on the targeted layer-by-layer printing frequency. Non-conductive substrates can be used when printing at sufficiently low frequencies or when charge dissipation is aided by a local injection of an ion flux or water vapor or by periodically discharging the substrate with a corona discharge needle[42]. Upon application of a voltage difference, typically in the 1000–2000 V range, a charged jet is expelled from the ink drop at the exit of the nozzle toward the substrate (Fig. 1a). The default trajectory of the jet is a straight line between the drop and the closest substrate point. The jet thickness is independent of the size of the nozzle aperture and it can be below ~100 nm depending on the ink properties, the supply rate and the applied voltage[43]. The viscoelastic properties of the jet need to be properly adjusted for it not to break into droplets. For this purpose, small amounts (2–10 wt%) of a polymer such as polyethylene oxide (PEO) with a high molecular weight, typically 300–5000 kDa, are usually incorporated into solution-based inks[43]. The separation between the nozzle and the printing substrate is typically in the range from 1 to 5 mm. Longer separations may result in whipping instabilities on the jet, which would hamper its precise localization on the substrate; whereas shorter distances may not allow enough time for the jet to sufficiently solidify or dry up before reaching the printing substrate. As will be shown later, under jet-deflection control shorter nozzle-to-substrate separations also limit the size of the printable area. Patterns are usually printed using a continuous jet. Eventually, this jet could be rapidly interrupted/restarted by electrostatically activating a gutter that collects the jet when its deposition onto the substrate is not desired[44,45].

We modified the conventional EHD printer architecture by setting additional electrodes around the jet (Fig. 1b, c). The purpose of these electrodes was to modify the electric field in the vicinity of the jet to deflect it from its default trajectory and control in this way its point of arrival at the substrate (Fig. 1d, e). The voltages at these electrodes were synchronized and produced by amplifying a computer-generated signal in a range from about −2000 V to about 2000 V. The movement and position of the XY mechanical stage supporting the printing substrate was also controlled and synchronized through the same computer. As in conventional 3D printing, the software had full control over the

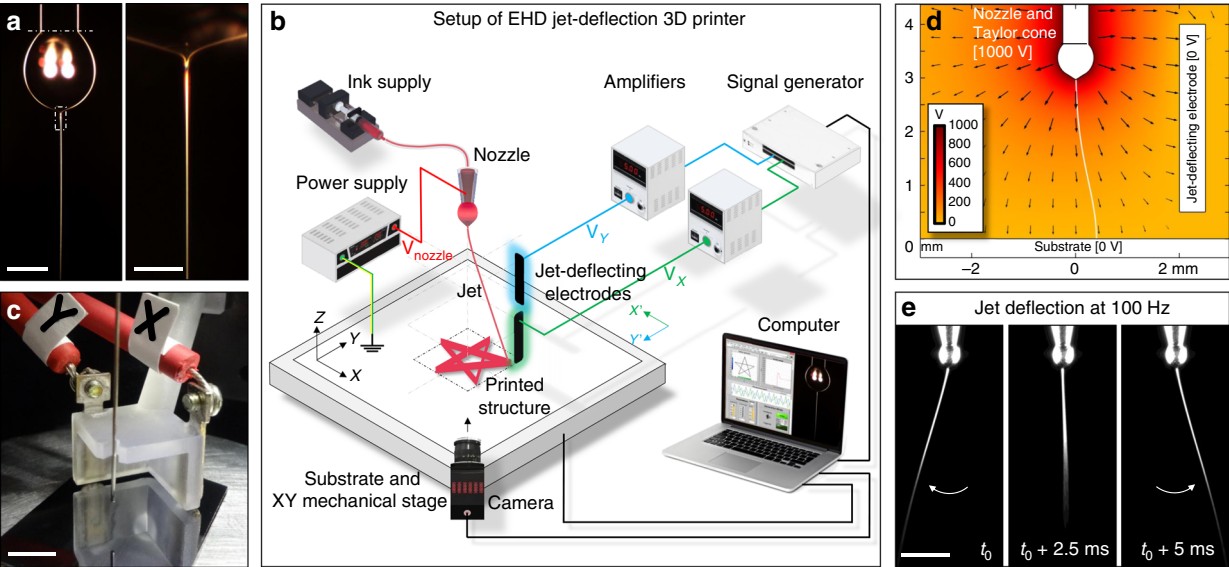

**Fig. 1 The electrostatic control of the jet trajectory. a** Optical photographs of the nozzle, ink drop (below dotted line), Taylor cone, and the electrified jet generated by applying 1000 V between the nozzle and a printing substrate (not shown). Scale bars: 500 µm and 50 µm. **b** Schematic of an EHD 3D printer with jet-deflecting electrodes. **c** Set of jet-deflecting electrodes and needle used as nozzle. Scale bar: 5 mm. **d** Simulation of the electric potential and field around the jet in the presence of a jet-deflecting electrode. The deflecting electrode, nozzle and ink drop are shown in white for clarity but are at the specified potentials. The electric field streamline (also in white) starting at the tip of Taylor cone represents the theoretical trajectory of a massless jet. Additional simulation results can be found in the supplementary material, Supplementary Fig. 1 and Supplementary Movie 3. **e** High-speed video captures of the jet being deflected in 1D with a frequency of 100 Hz (video showing jet deflection at 10, 50 and 100 Hz can be found in the supplementary material, Supplementary Movie 1). Two jet-deflecting electrodes (not shown) were used, positioned on the left and on the right side of the video captures. Video captures show the nozzle, the Taylor cone at the end of the ink drop and the thin jet expelled. The trajectory of this jet and thus its point of arrival to the substrate were modulated by the voltage applied to the jet-deflecting electrodes. Scale bar: 500 µm.

printing process through the parameterization of the layer-by-layer deposition process to print an object with predefined geometry, size and in our approach even microstructure.

**Pattern size, printing speed and jet deflection acceleration.**
Figure 1e shows high-speed video captures of a jet produced from an ethanol-water (3:1) ink containing 5% PEO ($M_v$ 300 kDa) being periodically oscillated from the action of a sawtooth wave applied at two opposing jet-deflecting electrodes (Supplementary Movie 1). This electrified jet remained stable under the action of the jet-deflecting field at frequencies as high as 10 kHz. Its deflection angle depended on the amplitude and frequency of variation of the electric field. For small signal amplitudes applied to the jet-deflecting electrodes, small deflection angles (<15°) that varied linearly with the voltage amplitude were observed. Higher voltage amplitudes eventually resulted in a non-linear increase in jet deflection angle, which was limited at around 40° before severe jet instabilities started to occur. At a low frequency, small oscillation amplitudes resulted in buckled fibers, while larger amplitudes produced straight fibers (Fig. 2a, b). As jet deflection frequency increased, the amplitude range resulting in buckling was decreased (Fig. 2c). As a result of this electrostatic jet deflection in one axis, fibers with a thickness down to 100 nm and pattern widths up to 2 mm could be deposited. Using at least two electrodes to deflect the jet in any direction along the substrate plane, 2D structures with any predefined shape could be produced (Fig. 3d, e). Placing the printing nozzle at 5 mm from the substrate and using moderate jet deflection angles to ensure a linear dependence with the voltage amplitude, the lateral size of the largest object defined by jet deflection was around 2 mm. Additionally, by coupling and synchronizing the jet deflection system with a mechanical stage, the fast printing of microscopic features could be extended to larger areas (Figs. 2, 3).

As the jet deflection frequency was increased, the jet deflection angle at a given amplitude decreased and the size of the printed pattern/object was reduced (Fig. 2c). This result was expected taking into account the finite speed of the electrified jet and the inability of the electrostatic deflection to appreciably stretch the jet. When attempting to print 2D and 3D patterns faster than the jet speed at its arrival to the substrate, the fiber length became insufficient to complete the intended pattern and thus degraded geometries were produced. To avoid this limitation, the jet deflection waveform had to be designed so that the contact point between the jet and the substrate was moving at all times at the same speed of the jet at its point of arrival to the substrate. For the printing speed to match the jet speed it was required to find the appropriate jet deflection parameters, which could be easily accomplished when using a calibration pattern such as those shown in Fig. 3b, c.

Additionally, calibration patterns produced by electrostatic jet deflection allowed to easily determine the speed of electrified jets, which is otherwise extremely challenging to measure due to their small width and high speed. The jet speed was measured as the length of fiber deposited per unit of time, which was easily determined from the macroscopic width of a printed pattern made with a repeating motif and with no fiber buckling, multiplied times the number of motifs printed per second, i.e. the known jet deflection frequency (Figs. 2, 3). The jet speed strongly depended on printing parameters such as the voltage applied between the nozzle and the substrate, the ink flow rate and the ink composition, which determined its electrical conductivity and viscosity. Analysis of the printed patterns revealed a highest jet speed of 0.5 m s$^{-1}$ when pulling an ethanol-water ink containing 5% PEO (300 kDa) with a voltage of 1200 V and a flow rate of 0.05 µl min$^{-1}$. This speed was one hundred-fold higher than the average speed of our fast mechanical stage following a back and forth trajectory with an amplitude of 10 µm (Supplementary Table 2).

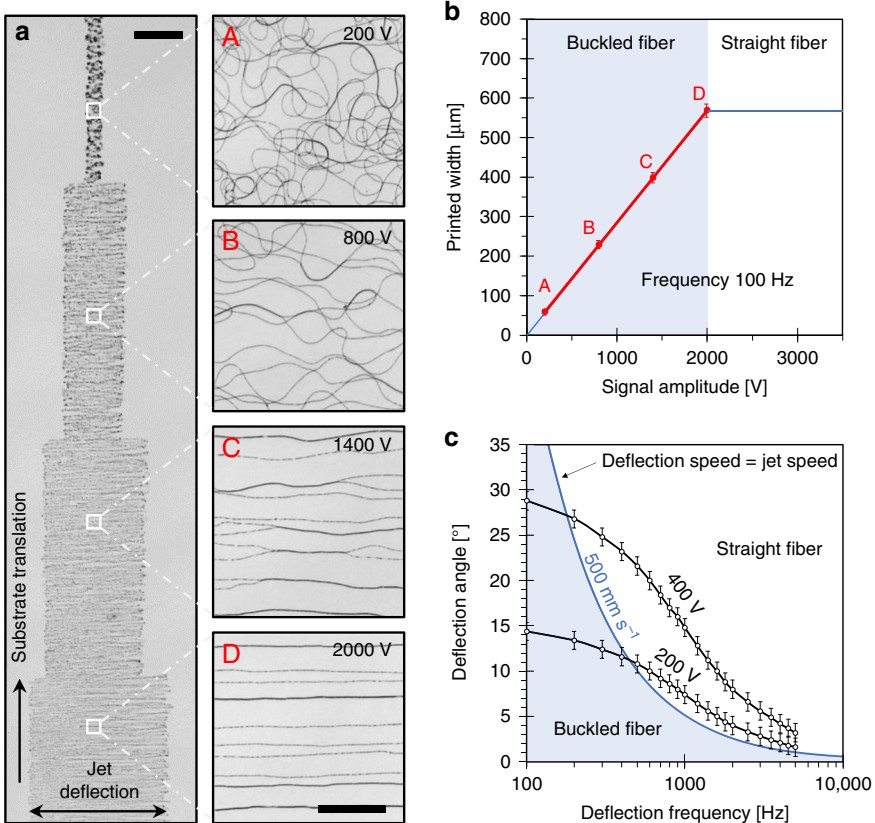

**Fig. 2 Role of jet deflection parameters. a** Optical photographs of the PEO fiber collected as the substrate is moved at 1 mm s$^{-1}$ and the jet is deflected with a frequency of 100 Hz. The stepwise increase of amplitude of the jet deflection signal resulted in a stepwise increase of width of the PEO pattern, fiber straightening and alignment. The amplitude of the jet deflection was varied from 200 V to 2000 V as depicted in the micrographs and the two electrodes were located at 10 mm from the default jet trajectory. Scale bars are 200 µm on main panel and 25 µm on magnified panels. **b** Dependence of the width of the printed pattern on the signal amplitude at a fixed frequency of 100 Hz. The blue shaded area displays the amplitude range where fiber buckling would be obtained at this fixed frequency. The four points in red correspond to the four amplitudes experimentally tested and presented in **a**. At the lowest amplitude that provides straight fibers, the printing speed and the jet speed are matched (point D) and the jet speed can be computed as a product of the fiber length printed in one deflection period times the printing frequency. At this amplitude, the width of the printed pattern reaches a plateau and it cannot be increased by increasing the signal amplitude. **c** Experimental dependence of the jet deflection angle on the jet deflection frequency for two different jet deflection amplitudes, 200 V and 400 V (two opposing electrodes located at 3 mm from the default jet trajectory). A blue line corresponding to a jet deflection speed of 0.5 m s$^{-1}$ is also plotted. The blued shaded area displays the region providing a jet deflection speed below 0.5 m s$^{-1}$, thus jets traveling at this speed would result in fiber buckling. Error bars were determined using the standard error of the mean of five or more measurements.

As noted above, high accelerations were essential to continuously match the in-plane printing speed, i.e., contact point speed, to the electrified jet speed when printing complex 2D and 3D patterns. From measurements of jet deflection angles at 10 kHz using a high-speed camera, we determined that lateral jet accelerations could reach well above 500 km s$^{-2}$ if they were not limited by the jet speed (Supplementary Note 1). This value is three to four orders of magnitude higher than those of fast precision mechanical stages, which are limited by their relatively large mass and the need for avoiding excessive vibration to maintain precision. Such huge accelerations allowed printing lines with a radius of curvature below 1 µm (Supplementary Fig. 2), although this value depended on the viscoelastic properties of the jet on its arrival to the substrate, as discussed below.

**Printing of 3D objects.** 3D objects were printed by the successive layer-by-layer deposition of material. Through this procedure, 3D structures with height up to 100 µm and very high aspect ratios, e.g. walls with height-to-thickness ratios well above 1000, were easily printed. Figure 4 displays a schematic diagram of this process, and several SEM micrographs of straight walls printed by

stacking up to 150 layers on top of each other. Using PEO inks with relatively low electrical conductivities, we were able to produce 3D structures using layer-by-layer frequencies as high as 2000 Hz. The fast jet and these high layer-by-layer frequencies translated into printing speeds up to 0.5 m s$^{-1}$ in-plane and 0.4 mm s$^{-1}$ off-plane, i.e., in the vertical direction, three to four orders of magnitude faster than achievable by extrusion and drop-on-demand EHD techniques when producing equivalent feature sizes. By either increasing the electrical conductivity of the printed material or using a proper atmosphere to allow faster charge dissipation, 3D printing speeds could be further increased.

To achieve efficient layer-by-layer printing, the electric charge carried by the electrified jet and remaining on the printed material must be dissipated fast enough to allow the next layer to assemble on its top. If charge was not dissipated fast enough, the new arriving jet containing charge of equal polarity was repelled, thus falling on random or imprecise locations. Taking into account the electrical conductivity of the inks (Supplementary Table 3) and the printed polymer, a charge relaxation time in the range $10^{-3}$–$10^{-6}$ s was estimated (Supplementary Note 3). Thus, for conventional inks based on PEO and using a conducting substrate electrically Earth-grounded, charge dissipation sets a

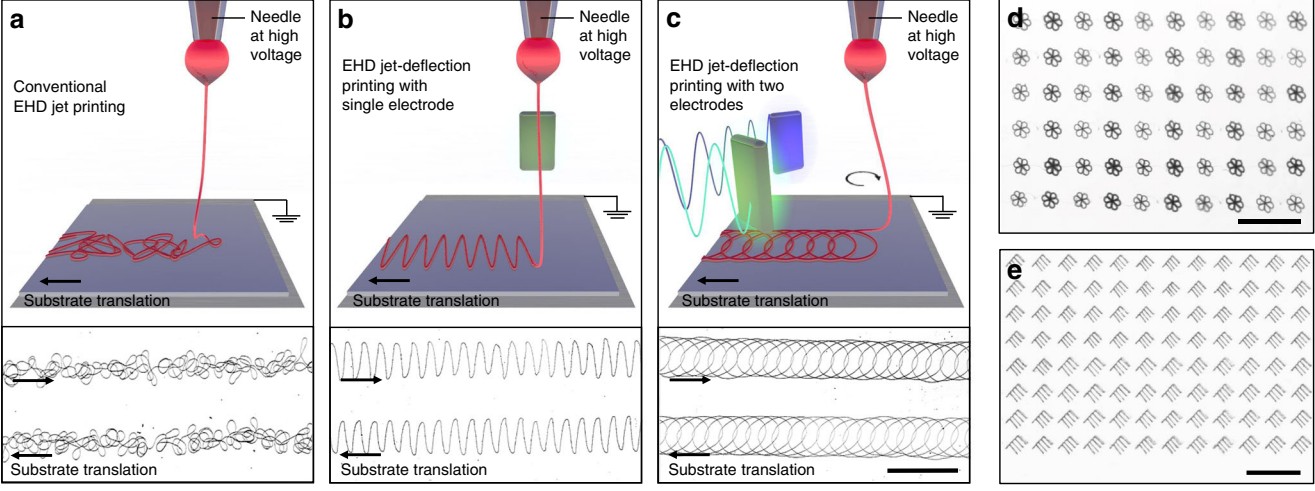

**Fig. 3 Printing 2D patterns. a–c** Schematics (top panels) and optical photographs of the experimental PEO-PEDOT:PSS patterns printed as the substrate is continuously moved in a straight line (bottom): **a** fiber buckling obtained with no jet deflection; **b** sawtooth pattern obtained using 1D jet deflection in an axis normal to the translation of the mechanical stage; and **c** circular pattern obtained using 2D jet deflection. All optical images have the same scale of 250 μm. **d, e** Optical photographs of more complex 2D patterns printed using two jet-deflecting electrodes to define the pattern and the mechanical stage to translate the substrate between printing events. A 4.7 wt% PEO ink containing Ag NPs was used to print these patterns. Scale bars (**d, e**): 1 mm.

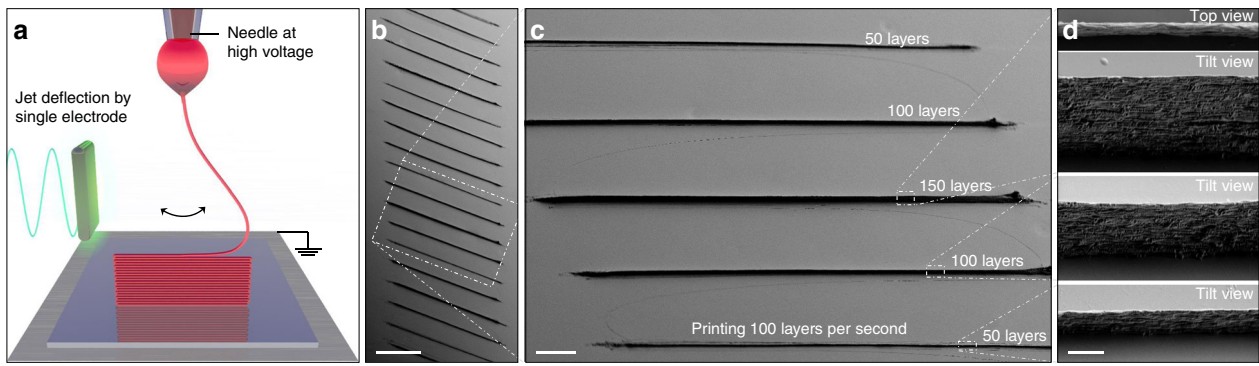

**Fig. 4 Printing 3D walls. a** Schematic of the 3D printing of a wall. **b–d** SEM micrographs of PEO walls built by layer-by-layer assembly at a jet oscillation frequency of 50 Hz, thus depositing two layers per period. Each wall was printed using exclusively electrostatic jet deflection to position the material on the substrate. The XY translation stage was moved only in between walls. The periodic deflection of the jet during 1.5, 1 and 0.5 s resulted in walls of variable height, composed of 150 layers, 100 layers and 50 layers, respectively. SEM micrographs in **d** shows the top view of the wall composed of 150 layers and the 45° tilt view of walls composed of 150, 100 and 50 layers. Scale bars (**b, c**): 200 μm and 50 μm. All SEM micrographs on **d** have the same scale of 2 μm.

limit to the maximum printing frequency at $10^3 - 10^6$ layers per second, depending on the wetness of the printed fiber.

After ensuring a fast-enough charge dissipation, the charged nature of the jets was actually highly convenient not only to achieve high printing speeds and accelerations, but also to easily and precisely manufacture 3D objects by self-assembly of the new arriving jet on top of a previously printed structure. Upon reaching the substrate, the electric charge carried by the jet is gradually dissipated by conduction to the substrate, and is replaced by charge of opposite polarity, as ohmic conduction through the printed fiber lowers its electric potential (towards 0 V, the potential of the earth-grounded substrate). This opposite-polarity charge accumulates at the top-most surface of the printed object, locally enhancing the electric field. Therefore, the newly arriving jet is electrostatically attracted to this charge, tending to self-assemble with high precision on top of the previously deposited layer. This attraction was fundamental to accurately accumulate layers, overcoming limitations associated to the movement-induced vibrations of mechanical stages[20]. This is the same mechanism termed electrostatic autofocusing by Galliker et al.[27] underlying the direct-printing of high-aspect-

ratio nanostructures using electrically-charged colloidal nanodroplets.

Considering that the height step of each layer was relatively small, <1 μm, compared to the distance between the nozzle and the substrate, ca. 5 mm, the contribution of the reverse charging to the electric field was extremely local. Thus, while facilitating layer-by-layer assembly, the confined perturbation of the electric field still allowed the printing of complex structures that involved fiber crossovers and line-to-line separations as small as ~1 μm. Figure 5 displays some examples of 3D structures produced from PEO and a combination of PEO and Ag nanoparticles using jet deflection. Even suspended filaments and structures could be produced when the jet was sufficiently dry upon reaching the printing substrate. These examples illustrate that the jet deflection strategy can probably be also used for printing on porous or non-planar substrates such as paper (Supplementary Fig. 4), pillars, meshes and curved surfaces.

The jet deflection mode could be combined with the stage translation to produce larger objects. Supplementary Fig. 3 displays SEM images of 10 mm straight walls produced by oscillating the jet in the same direction as the substrate is moved.

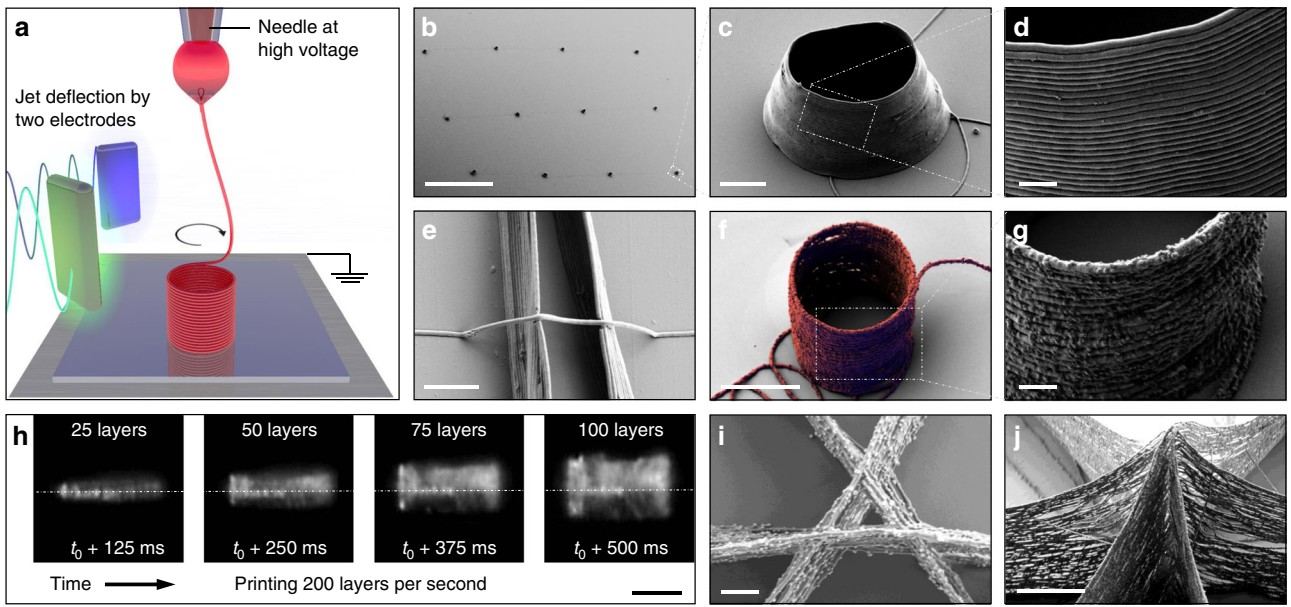

**Fig. 5 Printing 3D structures. a** Schematic of the 3D printing of a cylinder. **b–d** SEM micrographs at different magnifications of PEO 3D cylindrical microstructures manufactured by EHD jet deflection printing. Scale bars (**b–d**): 200 μm, 5 μm and 1 μm. **e** SEM micrograph of a single suspended PEO fiber bridging a gap between 2 parallel nanowalls. Scale bar: 2 μm. **f, g** SEM micrographs of a PEO-Ag cylindrical structure printed using an ink containing 5 wt% 50 nm Ag nanoparticles. Scale bars: 5 μm and 1 μm. **h** High-speed video captures displaying the growth of a cylindrical structure at a frequency of 200 Hz. The jet of PEO ink had a diameter of ca. 200 nm and it is invisible on these captures (Supplementary Movie 2). Scale bar: 20 μm. **i, j** SEM micrographs of the crossing of three walls printed using an ink containing 50 nm Ag nanoparticles, where (**i**) is a top view of a crossing having a gap of 1 μm and (**j**) is a tilt view of the peak formed by walls crossing in one point. Scale bars (**i, j**): 1 μm and 5 μm. SEM micrographs (**b–d, f, g, j**) were taken with a 40 degree tilt, **e** with 30 degrees tilt, and (**i**) with no tilt. High-speed video captures (**h**) were taken at a shallow angle to the substrate. Image (**f**) was obtained by superimposing two images taken with secondary electrons and in-lens detectors, where printed fiber was false-colored in red and blue, respectively.

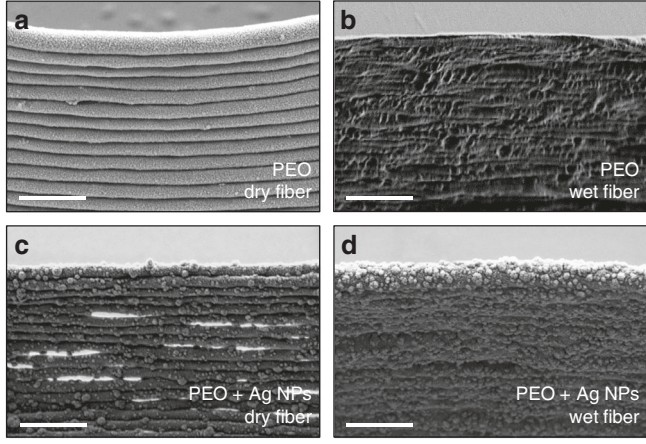

**Fig. 6 Control of the wall microstructure.** SEM micrograph showing the effect of jet wetness/viscosity upon arrival to the substrate on the microstructure of a printed wall: **a** PEO wall obtained by stacking dry-arriving fiber layers, where the different PEO layers are clearly distinguishable. **b** Compact PEO wall obtained by stacking wet-arriving fiber. **c** PEO-Ag wall produced from dry-arriving fiber, where layers and pores between them are clearly visible. **d** Compact PEO-Ag wall produced from wet-arriving fiber. All scale bars are 1 μm.

Large objects with *a priori* any predefined geometry can be potentially obtained by synchronizing the electrostatic jet deflection with the movement of the substrate on each printed layer, which requires the development of a proper 3D slicing software. The focusing effect may limit the similitude of a newly printed layer with respect to the previous one, i.e., very small differences of the printing angle, curvature and location from layer to layer may be prevented by the focusing effect. We estimate this effect to be dominant at a micrometer range distance, thus printing consecutive layer with differences on this order may require a different or additional strategy.

**Solvent evaporation.** In the additive manufacturing of 3D objects using solvent-based inks, the solvent evaporation rate is a fundamental parameter determining viscoelastic properties of the jet during fly and upon arrival to the substrate. Solvent evaporation rates must be low enough for the nozzle not to clog, but high enough for the jet to arrive sufficiently dry to the printing substrate. Within the wide range of ink viscosities enabling EHD 3D printing, smaller minimum curvature radiuses on the printed substrate were achieved with jets conserving a low viscosity on their arrival to the substrate (Supplementary Fig. 1b), whereas the printing of fiber bridges required jets reaching the substrate with higher viscosities (Fig. 5e). Additionally, once the material is printed, further solvent evaporation and associated volume loss may result in fiber shrinkage, which generally has a negative effect on the geometric fidelity of 3D structures. As an example, Fig. 5c shows the truncated cone that resulted from the drying of a straight wall cylinder similar to those shown in the high speed video captures of Fig. 5h (Supplementary Movie 2). By modifying the ink composition and adjusting environmental conditions, shrinking could be controlled and even avoided (Fig. 5f).

The solvent evaporation rate depends on multiple parameters, mainly the ink composition and particularly the equilibrium vapor pressure of the solvent used, the ambient conditions, the size and speed of the jet, and the diameter of the pendant drop. This last parameter, the diameter of the pendant drop, which depends on the needle size, the flow rate and the solvent evaporation among other parameters, has a strong influence on the printing process and particularly on jet deflection as noted in the supplementary material (Supplementary Note 2). Besides

adjusting the ink composition and tuning the printing parameters, to adjust the rate of solvent evaporation, the printer may be placed within a chamber with a controlled atmosphere or a gas stream containing solvent vapor can be introduced coaxially or near the jet. The control of the gas atmosphere may also be necessary to prevent that ambient moisture or oxygen gets absorbed by sensitive inks, which may cause phase separation or degradation of the ink[46].

The control of the drying process also allowed tuning the microstructure of the printed object. Smoother and less porous walls were obtained with jets that preserved a relatively low viscosity upon reaching the substrate, while jets arriving much dryer/solid resulted in rougher and highly porous walls (Fig. 6). This control of the microstructure opens avenues for applications where surface area and porosity of the printed features are key parameters, such as in catalysis, sensors, nanogenerators, microbatteries, and tissue engineering, to cite a few.

**Material versatility.** As mentioned before, material versatility is a key advantage of 3D printing based on the layer-by-layer deposition of material ejected from a nozzle over other technologies. While most of the printed structures displayed in the present work were produced from PEO (Figs. 2–6, Supplementary Table 1), structures printed from other polymers could be also produced by proper ink formulation. As an example, patterns displayed in Fig. 3a–c were obtained using a combination of PEO and PEDOT-PSS. Additionally, nanoparticles of any material can be incorporated into the ink. As an example, Fig. 5f, g, i, j display structures produced from an ink containing 5 wt% of Ag nanoparticles and 4.75% in mass of PEO. Besides, inorganic structures can be also produced from the printing of inks containing molecular precursors or metal salts and subsequently annealing them[23]. Ultimately, the range of printable materials is only constrained by the requirement that the ink has proper electrical conductivity and viscoelastic properties to flow and prevent its capillary breakup. Therefore, except for minor adjustments in formulation, the electrostatic jet deflection strategy can be extended to produce 3D objects from any of the materials that have already been made into fibers by electrospinning, including biomaterials and even living cells[47].

**Technology comparison.** Figure 7 shows a plot of the printing speed ($\mu m^3 s^{-1}$) as a function of the feature size (voxel size, μm) for additive manufacturing techniques capable of printing metals with submicron resolution. This plot is an adaptation from Hirt et al.[48]. The plot includes data for direct ink writing (DIW), drop-on-demand EHD printing, local electrophoretic deposition, laser-induced forward transfer (LIFT), meniscus-confined electroplating, electroplating of locally dispensed ions in liquid (FluidFM/SICM), laser-induced photoreduction and focused electron/ion beam induced deposition (FEBID/FIBID)[48]. As a general trend, when manufacturing objects with smaller features the printing speed drastically decreases; each order of magnitude increase in printing resolution results in 4 orders of magnitude slower printing. In this context, the electrostatic jet deflection strategy presented here falls well above the general trend, being capable of printing feature sizes down to 100 nm with unprecedented printing speeds up to $10^5 \mu m^3 s^{-1}$.

## Discussion

In conclusion, we presented a strategy to enable a fast printing process in nozzle-based 3D printing techniques, based on the control of the trajectory of an electrified jet by means of rapidly tuning the surrounding electrostatic field through additional electrodes. EHD jetting allows producing submicrometer fibers

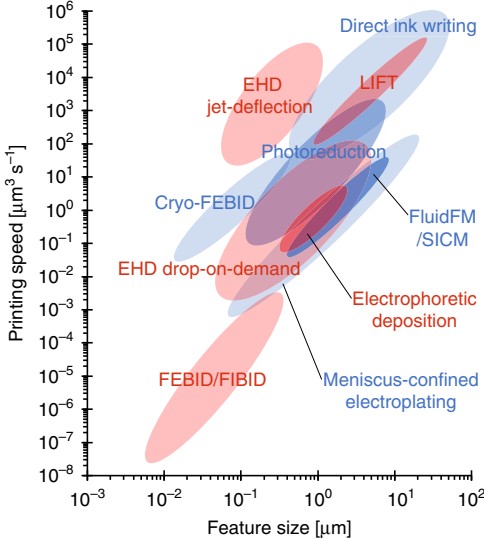

**Fig. 7 Comparison of additive manufacturing technologies.** Map of printing capabilities in terms of printing speed and feature size for manufacturing technologies providing submicron resolution. Adapted from Hirt et al. and extended to include EHD printing based on jet deflection[48].

from wide nozzles and with a broad range of ink viscosities. By coupling additional electrodes to this system, ultrafast dynamic characteristics were reached. We demonstrated printing speeds up to 0.5 m s$^{-1}$ in-plane and 0.4 mm s$^{-1}$ off-plane, surpassing all known additive manufacturing techniques capable of providing submicron resolution. Additionally, accelerations up to $10^6$ m s$^{-2}$ were calculated, four orders of magnitude above those provided by techniques relying on mechanical stages to define the object geometry. Such high accelerations allowed printing patterns with radius of curvature down to 1 μm. Through electrostatic deflection of electrified jets, 3D structures of increasing complexity, including crossovers and bridges, were printed by precise electrostatically-driven layer-by-layer self-assembly at frequencies as high as 2000 layers per second. Besides, controlling the ink viscosity and composition allowed adjusting the microstructure of the printed objects. To sum up, we believe that the advantages of EHD jet deflection printing will represent a significant step forward toward ultrafast additive micromanufacturing of 3D objects with virtually any composition and adjusted microstructure and functionality.

## Methods

**Chemicals.** Polyethylene oxide (PEO) of various molecular weights was purchased from Sigma-Aldrich (#182001, viscosity-average molecular weight 300 kDa; #372781, 1000 kDa; #189472, 5000 kDa). Poly(3,4-ethylenedioxythiophene) polystyrene sulfonate (PEDOT:PSS) solutions were purchased from Sigma-Aldrich (#655201, 3–4% in water). Ethanol and ethylene glycol were obtained from different sources. All chemicals were used as received, without further purification. Ag nanoparticles of ca. 50 nm in diameter were synthesized using PVP as ligand[49]. After synthesis, nanoparticles were thoroughly washed by multiple precipitation and redispersion cycles using ethanol as solvent and acetone as antisolvent. Finally, nanoparticles were precipitated for posterior use. Thus obtained Ag nanoparticles could be dispersed in polar solvents such as water and ethanol.

**Inks formulation.** Deionized water was used as main solvent and some amounts of ethanol or ethylene glycol were added to control surface tension and evaporation rate. All ink compositions are described in Supplementary Table 1. PEO inks were prepared by dissolving the proper mass of PEO (typically 2–10 wt%) in the solvent mixture during 24 h under magnetic stirring. PEDOT:PSS inks were prepared by adding the proper amount of PEDOT:PSS solution into the dissolved PEO ink and homogenizing the mixture by magnetic stirring. Ag nanoparticle inks were prepared by adding the PEO ink into a flask containing precipitated Ag nanoparticles and dispersing them using ultrasonication and magnetic stirring. All inks were kept in sealed vials, where they could be stored for months and even years without

showing signs of degradation. Inks containing PEDOT:PSS were stored in a fridge at 4 °C.

**Printer set-up and printing process**. Inks were loaded into a glass syringe (Hamilton #81320, 1 ml) and supplied to the nozzle by a syringe pump (Harvard apparatus, Pump 11 Pico Plus Elite 70–4506) with a typical flow rate of 0.05–0.07 μl min⁻¹. The pendant drop formed at the nozzle aperture had a diameter of ca. 100–1000 μm. Stainless needles with blunt ending (Hamilton N726S, 26 s gauge, 127 μm ID, 474 μm OD) or borosilicate glass tips (c.a. 70–100 μm) were used as nozzle. No surface treatment was applied to the tips before their use. Glass tips were manufactured from borosilicate capillaries (Sutter Instruments, B100-50-15) by a Pipette puller (Sutter Instruments P-97) and the tips were manually broken by scratching two tips against each other, leaving glass tips with an outer diameter of c.a. 70–100 μm. Glass tips were glued atop of a stainless-steel needle with blunt ending (B Braun Sterican, 27 gauge). Silicon wafers (University Wafers #452, p-type, <100>) were used as substrate. Silicon was cleaned with isopropanol to remove organic contamination prior to printing. The nozzle-to-substrate distance ranged 2–5 mm, but typically was 3 mm. A positive potential was applied to the nozzle using a Matsusada AU-20P15, max. 20 kV, 15 mA high voltage supply, while the substrate was positioned on an electrically grounded plate, mounted atop of a XY translation stage (PI miCos linear stages PLS-85 with 10 mm range in both X and Y with RS422 encoders).

Electrohydrodynamic (EHD) jetting was initiated by slowly increasing the nozzle voltage up to 1800–3000 V, until the pendant ink drop elongated and fell on the printing substrate. Jet initiation voltage was lower for inks with lower surface tension (those containing ethanol), and also for nozzles and pendant drops of smaller size. Once the jet was initiated, voltage was lowered to 400–1500 V and the jet was stabilized for 2 min before printing. Lowest nozzle voltages in this range were used for inks containing PEO with very high molecular weight (i.e., 5000 kDa), which provided superior jet stability due to their higher viscoelasticity. Jet stability was critical for printing reproducible features and it depended on printing parameters such as the applied voltage, ink properties including viscosity and volatility, and on ambient parameters that together with the flow rate determined the size and stability of the drop. A proper ink formulation and selection of the printing parameters, including voltages and flow rate at the syringe pump, provided jets that remained stable for over 15 min, which was the time period of the longest printing event we studied. Considering that single objects were printed in less than 1 s, 15 min allowed printing over 1000 objects.

The printing process was carried out under ambient conditions, with temperature in the range 18–25 °C and 40–80% relative humidity. During printing the substrate was translated relative to the nozzle by the XY translation stage. Translation was either continuous (to produce fiber tracks, as on Fig. 3a–c) or in steps (to print 3D patterns while the substrate was motionless, as on Figs. 4, 5).

The jet was deflected from its default trajectory using jet-deflecting electrodes. For the high-speed video study of the jet deflection and for Fig. 2c, two electrodes were positioned on the same axis (left and right on Fig. 1e) and the separation between the default jet trajectory and electrodes was 3 mm. A fixed electrodes design (as on Fig. 1c) was used for EHD printing, with a nozzle-to-electrodes separation of 10 mm. Electrodes were glued to a plastic holder produced by polymerization of photosensitive resin (Formlabs Form 2 3D printer, FLGPCL04 clear resin).

A LabVIEW software was specifically developed to define and generate the voltage at the different jet-deflecting electrodes from parameters such as the pattern geometry, the number of printed layers, the layer printing frequency and the signal amplitude. A data acquisition card (National Instruments, USB-6259) was used to generate the synchronized analog signals (max. ±10 V), which were amplified (max. ±2000 V, Matsusada AMJ-2B10 and Trek 677B) and applied to the jet-deflecting electrodes. The amplitude of the signals applied to jet-deflecting electrodes defined the size of the printed pattern and typically ranged about 1000–2000 V for 10 mm nozzle-to-electrodes separation and 200–400 V for 3 mm nozzle-to-electrodes separation. Printing frequencies typically ranged from 50 to several hundred layers per second, but we could reach layer-by-layer frequencies as high as 2000 Hz. The optical microscope used to monitor the printing process consisted of a 12X lens with adjustable zoom and focus (Navitar 1-50486), a 2x lens adaptor (Navitar 1-62136) and a 5X microscope lens (Mitutoyo 1-60226). A high-intensity light source (AmScope HL-250-A) was positioned behind the observed object under ca. 5° angle from the camera optical axis, thus that direct light did not reach the camera sensor (dark-field setting, as on Figs. 1a, e, 5h). For real time monitoring of the printing process, a paper sheet was put between the light source and the observed object (bright-field setting). Printing process was monitored and recorded using a CMOS camera (Basler acA2040-25gc) mounted on the microscope. High-speed videos (Supplementary Movie 1, captures on Fig. 1e, and Supplementary Movie 2, captures on Fig. 5h) were recorded with a high-speed camera (Photron FASTCAM-1024PCI) at 1000 fps. Most of printed samples were sputtered with a thin silver layer using a DC magnetron sputter (Emitech K575X, 80 mA, argon, 90 s, sample was slowly rotated to obtain uniform thickness) to improve the quality of SEM images and to protect the PEO fiber from degradation/shrinkage caused by the electron beam. Scanning electron microscopy (SEM) micrographs were obtained at 1–2 kV electrons acceleration voltage on AURIGA (FIB-FESEM) from Carl Zeiss using a secondary electrons (SE2) detector. Figure 5f was obtained by superimposing two images taken with SE2 and in-lens detectors and false-coloring the printed object in red and blue respectively.

The simulation of the electric potential and field around the jet in the presence of a jet-deflecting electrode (Fig. 1d) was done in COMSOL using the following parameters: nozzle potential at +1000 V; substrate and jet-deflecting electrode electrically grounded (0 V); nozzle-to-substrate separation of 3.6 mm; nozzle-to-electrode separation at 2 mm, and electrode size (height, width and thickness): $3.5 \times 3.0 \times 0.5$ mm³. Color gradient represents the electric equipotential lines and black arrows represent the electrical field vectors and are proportional to its magnitude. The deflecting electrode, nozzle and ink drop were plotted in white for clarity, but are at the specified potentials. The electric field streamline (also in white) starting at the tip of Taylor cone represented the trajectory of an electrostatically deflected massless jet. Supplementary Movie 3 presents an animation of jet deflection simulation using 2 jet-deflecting electrodes.

## Data availability

The dataset generated during and/or analyzed during the current study are available from the corresponding author on reasonable request.

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

## Acknowledgements

This work was supported by the Spanish Ministerio de Economía y Competitividad through the project SEHTOP (ENE2016-77798-C4-3-R) and Generalitat de Catalunya through the projects 2017SGR1246 and 2017SGR1516. IL acknowledges financial support from Generalitat de Catalunya (grant 2019FI_B2_00214).

## Author contributions

I.L., J.R.L. and A.C. devised the concept and designed the study. J.R.L. and A.C. supervised the project and I.L.'s contribution to the work. I.L. built the printing setup and wrote the printing software. I.L. designed and performed printing experiments reported here. I.L. performed SEM and optical microscope analysis. The work builds on results of I.L.'s PhD thesis. All authors discussed the results. A.C. and I.L. wrote the original paper draft and I.L. visualized the data. All authors reviewed and commented on the manuscript.

## Competing interests

The authors declare no competing interests.
