## [Peer Review File · Nature Communications]

Reviewers' comments:

Reviewer #1 (Remarks to the Author):

This manuscript reports a potentially ultrafast 3D printing method to fabricate submicron features using electrostatic jet deflection, enabled by tuning the surrounding electric field through additional electrodes. Depending on the electric field and printing process, the jets oscillate at a very fast speed with various deflection angles, which leads to ultrafast deposition process. This is an interesting idea built upon near-field electrospinning. But it needs more in-depth work and discussion to strengthen the conclusions.

Major points:

The manuscript does not include sufficient study on the physics how to control the electrified jet to precisely deposit onto the substrate. Fig. 1d shows deflection of the jet, but with deflection voltage of zero. There are no further simulation results and discussion on the jet deflection controlled by the deflection voltage, electric field between the nozzle and the substrate, ink properties, etc. How stable is the jet during printing? How does the droplet size affect the jet formation, jet deflection, and deposition? In the supporting videos, the droplets (or meniscus) at the tip of the nozzle are different at 10 Hz, 50 and 100 Hz. From literature, the droplet size does affect the jet behavior. [Ref. He, X.; etc. *Journal of Physical Chemistry C*, 2017, 121, 8663-8678; Shin, D.; etc. *Journal of Micromechanics and Microengineering*, 2019, 29(4)]

What if the mechanical stage moves in the same direction as the deflection jets? It seems like when printing objects larger than the jet deflection width, the mechanical stage will need to move in the same direction as the deflection jets. What is the print process to obtain Fig. 3d and 3e? Line 106-115, the authors briefly described the 3D printing process of the proposed method, by synchronizing the voltage application on the deflection electrodes and mechanical stage movement. However, the reviewer cannot see a clear path for the realization of printing 3D objects with arbitrary shapes and structures, especially when the mechanical stage moves in the direction where the deflection occurs. In addition, it seems that all the printed objects are demonstrated with one continuous jet. Would this be the printing mode for larger objects too, or on/off of the jet will be controlled within one printed structure? What if an object needs to be printed with a curvature on the 1st to 5th layers of 1 micron, and on the 6th layer with another curvature or no curvature? How to overcome the focusing effect by the printed layers?

Minor points:

Is the jet deflection speed in Line 176 the same as the jet speed in Line 185?

Line 242, the role of image charges has not been well articulated. It is basically the auto focusing effect described in Ref. 27, i.e., the jet is attracted along the shortest electrical path if the charges on the printed layers are dissipated sufficiently fast.

Line 236, how to get the charge dissipation time of 10^{-4} to 10^{-6} s? What is the conductivity of the ink?

The term of off-plane is confusing. Does the off-plane refer to the Z-axis?

This work presents an interesting method to quickly generate fine structures, and it definitely has some potential applications, for example, tissue engineering, electrical interconnects, etc. But the reviewer thinks that it is too strong a claim as a 3D printing method, especially when directly compared with conventional 3D printing.

Reviewer #2 (Remarks to the Author):

This manuscript reports on the use of electrohydrodynamic jetting to print 2D and 3D features. The key contribution of this work is the use steering electrodes (i.e. electrodes positioned around the emitted jet) to deflect the jet trajectory at high speed. While past work has used EHD jets to

print materials - these previous efforts have generally relied on the use of mechanical stages to create 2D/3D features (which are inherently slow). This work reports on a new capability that combines jet deflection (and the use of stages) to increase printing speed. The authors show that this technique can also exert control over the nanostructure of the printed features by controlling the evaporation rate of the solvent in the jet.

I believe this is good quality work and a novel contribution. The manuscript is generally well-written. The following items should be addressed before consideration for publication.

1. One of the unique features of this technology is the "self aligning" aspect of one layer to the next when printing 3D structures (described on Page 11). While this certainly exists and is reported by others - I am confused by the explanation given in the manuscript. In particular, the use of the term "image charge". From electrostatics, I believe image charge refers to a "fictitious" charge used to solve for a field (for example, the classic solution to the point-plane problem.) In the manuscript - image charge is referring to a charge of opposite sign (to the applied potential, potential of fiber) that collects at the top of the printed features. But why should this happen? The printed fibers should initially have the same potential as the "newly arriving" fiber. This charge on the printed fibers should rapidly dissipate - but it is then replaced with charge of the opposite polarity? Perhaps it is better to assume that there is electrical continuity from the top of the printed features (to ground)? And if this is the case - would the fibers need to behave as conductors? But if they are polymers - do they not act more like (leaky?) dielectrics? Perhaps all of this can be clarified - as I think it is an important aspect of this work.

2. Following from point 1 above - is it not required that the target substrate be conductive (or at least a semiconductor) in order to print onto it? The initial charge on the printed fiber must be dissipated. This would imply that printing on insulating regions would not be possible (it is suggested it would be possible on page 12).

3. For this printing methodology to work, the contact point between the jet and substrate and the jet speed must be equal. Surely achieving this balance must be difficult. The jet speed and deflection must (independently) be related to a number of factors (conductivity of solution, applied potential, etc.). But the dependence of jet speed and deflection on these factors is only briefly addressed (this should be expanded on). In addition - can a statement be made on how difficult it is to equate the jet speed and contact point speed for printing.

4. A minor point: what would be the largest feature that can reliably be printed? I understand this can likely be calculated from the separation distance and deflection angle - but an explicit statement may be appropriate.

5. While this technology has the advantage of speed - one potential disadvantage over other printing technologies is the apparent inability to print sacrificial materials (for example, to print interior features, etc.) Certainly this is an important capability of 3D printing. Can the authors address this point?

Reviewers' comments:

Reviewer #1 (Remarks to the Author):

This manuscript reports a potentially ultrafast 3D printing method to fabricate submicron features using electrostatic jet deflection, enabled by tuning the surrounding electric field through additional electrodes. Depending on the electric field and printing process, the jets oscillate at a very fast speed with various deflection angles, which leads to ultrafast deposition process. This is an interesting idea built upon near-field electrospinning. But it needs more in-depth work and discussion to strengthen the conclusions.

Major points:

The manuscript does not include sufficient study on the physics how to control the electrified jet to precisely deposit onto the substrate. Fig. 1d shows deflection of the jet, but with deflection voltage of zero. There are no further simulation results and discussion on the jet deflection controlled by the deflection voltage, electric field between the nozzle and the substrate, ink properties, etc.

We thank the reviewer for pointing out his/her concerns on the insufficient discussion of the printing mechanism. Taking into account these concerns, we have corrected the manuscript in the following ways:

1. We added a series of results from finite element analysis of the electric field around the jet at different electrode potentials. These simulations were added as a new figure in the supporting information (Supplementary Figure 1).
2. We also added a movie including finite element analysis under different jet deflection voltages as Supplementary Movie 3.
3. Supported on these simulations, we extended our discussion on the influence of the electric field on the jet trajectory, considering parameters such as the deflection amplitude, voltage bias, number of electrodes and needle voltage. We introduced this extended discussion on the supplementary information, under the section Supplementary Figure 1.

Supplementary Figure 1. Simulation of the electric potential and field around the jet in the presence of one and two jet-deflecting electrodes. The nozzle (at the center), the ink drop (hanging at the tip of the nozzle), and the deflecting electrode are shown in white for clarity but are at the specified potentials. The electric field "streamline" (also in white) starting at the tip of Taylor cone (ink drop conical end) represents the theoretical trajectory of a massless jet. The jet deflections on the collector are written in blue.

Supplementary Figure 1 displays a series of simulations of the electric field around the jet at different electrode potentials and considering one and two jet-deflecting electrodes. Five

simulation sets were obtained considering different number of jet deflecting electrodes and voltages. Within each set a different parameter was studied.

Set 1 shows jet deflection using one electrode. Voltage bias (+70 V in this case) is required to keep the jet in vertical position. Jet is attracted and repelled from the electrode by applying signal amplitude of $\pm 90V$ relative to the bias voltage. The jet deflection distance is greater when the jet is attracted than when it is repelled, 500 μm compared to 300 μm . To correct for this effect and avoid distortion in the printing of predefined objects, our software is designed to dynamically decrease the amplitude when the jet is attracted, and increase it when it is repelled.

Set 2 shows configuration with 2 electrodes. As deflection signal amplitude is increased to $\Delta V=50$ V and $\Delta V=100$ V, jet deflection distance is linearly increased. This simulation result is supported by Fig. 2a-b in the manuscript.

Set 3 shows the effect of needle voltage by comparing to Set 2. The deflection distance is reduced proportionally to the increase in needle voltage ($\frac{1500\text{ V}}{1000\text{ V}} = \frac{450\ \mu\text{m}}{300\ \mu\text{m}} = 1.5$).

Set 4 shows how in a 2-electrode configuration voltage bias influences the jet deflection. While the amplitude of the deflection voltage is the same ($\Delta V=100$ V) for all pictures, the jet is deflected less as the bias increases. Stated differently, increasing the bias has a “focusing effect”, which decreases jet deflection.

Set 5 shows how drop size changes jet deflection. Simulation shows that a smaller drop results in greater jet deflection. Partially this effect can be attributed to a bigger separation between jet ejection point and substrate. However, the bigger cause of the effect arises from the perturbation of the electric field by the drop itself, i.e. smaller drop perturbs electric field around less (smaller area with red color).

How stable is the jet during printing?

Jet stability was critical for printing reproducible features as those shown in figures 3-5. Jet stability depended on the applied voltage, ink properties, particularly viscosity and volatility, and on ink flow rate and ambient conditions, which determined changes in the drop size. Thus, ink formulations were optimized to maximize jet stability. The jet was initiated by applying a voltage in the range between 1500 V and 3000 V. When a jet was generated, the voltage was reduced to 800-1500V. Then we waited for the jet to stabilize for 2 minutes. The jet remained stable for over 15 minutes, which was the time period of the longest printing event we used. Considering that a single object was printed in less than 1 second, 15 minutes allowed printing over 1000 objects. We introduced a comment in this direction in the experimental section of the manuscript, page 21. It should be noted in any case, that this jet stability does not depend on the jet deflection mode, but it is intrinsic of EHD jetting systems used for conventional electrospinning and near field electrospinning.

How does the droplet size affect the jet formation, jet deflection, and deposition? In the supporting videos, the droplets (or meniscus) at the tip of the nozzle are different at 10 Hz, 50 and 100 Hz. From literature, the droplet size does affect the jet behavior. [Ref. He, X.; etc. Journal of Physical Chemistry C, 2017, 121, 8663-8678; Shin, D.; etc. Journal of Micromechanics and Microengineering, 2019, 29(4)]

The drop size affects printing in different, direct and indirect ways. We experimentally found that smaller drops required smaller voltage applied to the needle to initiate the jet. This experimental observation can be rationalized considering that, at equal voltage applied, a higher electric field and a higher field gradient are generated on the surface and at the surrounding of smaller drops. Results of finite element analysis of the electric field with different drop sizes support this view (Supplementary Figure 1).

On the other hand, smaller drops resulted in slower solvent evaporation rates, which resulted at the site of ejection of the EHD jet in a lower polymer concentration than larger drops. The main consequence of this is that smaller drops lead to a more easily deformable (less viscous and more elastic) ink at the jet, thus a faster EHD jet (for the same electric field strength, despite a lower voltage is used). The drop size thus also influences the viscoelastic properties of the jet on its arrival to the substrate. Larger drops result in a more viscous jet arriving to the substrate and thus produce a more porous final 3D solid structure since different layers do not fuse together. The results obtained from the deposition of jets with variable dryness is displayed in Fig. 6 and Supplementary Figure 2. Additionally, solute enrichment at the interface of the drop can cause changes in the surface tension. A reduction in surface tension will result in a reduction in the required voltage needed to sustain the jet. In this case, the same deflection will be obtained with a lower deflection signal amplitude (ΔV).

Additionally, the effect of drop size on the electrical field causing the jet to deflect was further investigated via finite element analysis (Supplementary Figure 1). Finite element analysis shows that large drops substantially decrease the jet deflection distance. This result is explained considering two dependences: i) For small droplets, the vertical electric field strength decays faster as the jet moves away from the drop. As the vertical component of the electric field is weaker, the jet-deflecting field (horizontal field component) becomes relatively stronger, thus resulting in larger horizontal deflection of the jet. ii) While the nozzle elevation is kept constant, the jet ejection point is lower for larger drops. Thus, even at equal deflection angle, the jet contact point at the substrate will be moved a shorter horizontal distance.

We introduced this discussion as Supplementary Note 2.

What if the mechanical stage moves in the same direction as the deflection jets? It seems like when printing objects larger than the jet deflection width, the mechanical stage will need to move in the same direction as the deflection jets.

In the present work, to demonstrate the potential and reliability of the system, we have mainly focused on the printing of small objects using jet deflection to define the object shape and making use of the stage to move the substrate in between jet-deflection printing events. Using a nozzle-to-substrate distance of around 5 mm, the largest object that can be printed by only jet deflection is around 2x2 mm in size. Larger objects would require larger nozzle-to-substrate distances or the combination of the translation of the mechanical stage with the jet deflection. As a simple example of the combination of both movements in the same object, we printed straight walls while the substrate was moved along jet deflection plane. In this case, the length of the wall was around 10 mm, the maximum translation length of our mechanical stage. We introduced these data as a new figure Supplementary Figure 3. Generally, when combining the movement of the substrate with jet deflection to print more complex objects, both movements need to be synchronized to reproduce a predefined object. Although we have not explored this yet, a 3D slicing software should readily be able to account for both movements simultaneously and, accordingly, define the jet deflection voltage and movement of the translation stage to create a large 3D object. We have now introduced this discussion on page 13 of the manuscript.

What is the print process to obtain Fig. 3d and 3e?

To print figures 3d and 3e, the mechanical stage was moved in between each object and remained static while printing each object. The object size and geometry were defined by the jet deflection signals. The location of the objects on the substrate and the distance between them were defined by the movement of the mechanical stage. We have now clarified this point in the caption of figure 3.

Line 106-115, the authors briefly described the 3D printing process of the proposed method, by synchronizing the voltage application on the deflection electrodes and mechanical stage movement. However, the reviewer cannot see a clear path for the realization of printing 3D objects with arbitrary shapes and structures, especially when the mechanical stage moves in the direction where the deflection occurs.

Our software is currently designed to either control the voltage at the jet deflecting electrodes to print an object and then move the stage to print other objects sequentially (e.g. Figure 3d-e), or to continuously move the stage while the jet is deflected periodically to print a pattern with a repeating motif (e.g. Figure 3b-c). As noted above, to produce larger objects additional software development is needed to simultaneously combine jet deflection and stage movement. We foresee no limitation in defining such a software. Jet deflection being much faster than stage movement, the software must control mechanical stage and simultaneously update the direction of jet deflection to build objects with a predefined shape. We introduced a comment on this direction in page 13 of the manuscript.

In addition, it seems that all the printed objects are demonstrated with one continuous jet. Would this be the printing mode for larger objects too, or on/off of the jet will be controlled within one printed structure?

Currently the printer uses a continuous jet to produce the predefined object(s). Jet deposition can potentially be interrupted/restarted at will. One way to do so is by activating/deactivating a gutter that collects the jet when its deposition onto the substrate is not desired. The gutter is electrostatically controlled, which allows a very fast activation. Although such gutter was not used in the present work, whose primary goal was to print small objects at high speed, the ability to interrupt printing would be a useful future improvement. We introduced a comment on this direction on page 5 of the manuscript.

What if an object needs to be printed with a curvature on the 1st to 5th layers of 1 micron, and on the 6th layer with another curvature or no curvature? How to overcome the focusing effect by the printed layers?

We demonstrate in figure 5 that different layers can be printed on top of each other overcoming the very local focusing effect. However, we agree with the reviewer that this focusing effect currently limits the similitude of a newly printed layer with respect to the previous one. Therefore, very small differences from layer to layer of the printing angle, curvature and location may be disallowed by the focusing effect. We estimate the focusing effect to be dominant at a micrometer range distance, thus printing consecutive layers with pre-designed differences on this length scale will require a different/additional strategy or a proper electrostatic correction. This discussion was included in page 13 of the manuscript.

Minor points:

Is the jet deflection speed in Line 176 the same as the jet speed in Line 185?

Line 176 refers to the jet deflection speed, i.e. the speed of the jet contact point on the substrate. Line 185 refers to the jet speed in its translation from the nozzle to the substrate. As pointed out in the manuscript, both speeds need to be approximately the same to prevent fiber buckling/accumulation (when jet deflection speed is lower than the jet speed) or cutting corners (when the jet deflection speed is higher than the jet speed). This discussion can be found in page 8 of the manuscript.

Line 242, the role of image charges has not been well articulated. It is basically the auto focusing effect described in Ref. 27, i.e., the jet is attracted along the shortest electrical path if the charges on the printed layers are dissipated sufficiently fast.

In view of the comments of the two reviewers, we realize that the description of the focusing effect using the microscopic concept of image charges was not well articulated in the manuscript. Therefore, we removed the term "image charges" from the manuscript, which

now describes in detail the mechanism of charge conduction leading to polarity reversal on the already printed structure and electrical field enhancement towards such structure. This effect was referred as "electrostatic field-focusing" in Ref. 27, leading to the effect of an 'electrostatic autofocusing' of nanodroplets. We have now incorporated such terminology and reference to the manuscript. The new text including line 242 now reads:

After ensuring a fast-enough charge dissipation, the charged nature of the jets was actually highly convenient not only to achieve high printing speeds and accelerations, but also to easily and precisely manufacture 3D objects by self-assembly of the new arriving jet on top of a previously printed structure. Upon reaching the substrate, the electric charge carried by the jet is gradually dissipated by conduction to the substrate and is replaced by charge of opposite polarity, as ohmic conduction through the printed fiber lowers its electric potential (towards 0 V, the potential of the earth-grounded substrate). This opposite-polarity charge accumulates at the top-most surface of the printed object, locally enhancing the electric field. Therefore, the newly arriving jet is electrostatically attracted to this charge, tending to self-assemble with high precision on top of the previously deposited layer. This attraction was fundamental to accurately accumulate layers, overcoming limitations associated to the movement-induced vibrations of mechanical stages²⁰. This is the same mechanism termed "electrostatic autofocusing" by Galliker et al.²⁷ underlying the direct-printing of high-aspect-ratio nanostructures using electrically-charged colloidal nanodroplets.

In addition, we have removed the term previously appearing also in line 362. The new text reads:

Though electrostatic deflection of electrified jets, 3D structures of increasing complexity, including cross overs and bridges, were printed by precise electrostatically-driven layer-by-layer self-assembly at frequencies as high as 2000 layers per second.

Line 236, how to get the charge dissipation time of 10⁻⁴ to 10⁻⁶ s? What is the conductivity of the ink?

The charge dissipation happens by electrical conduction through the printed material to the conducting substrate. The charge on the surface of the printed material attains the equilibrium distribution in a time which is proportional to the so-called charge relaxation time. The charge relaxation time (τ) of a material can be computed from its dielectric permittivity (ϵ) and electrical conductivity (σ) (see for example Melcher, J. R. & Taylor, G. I. *Electrohydrodynamics: A Review of the Role of Interfacial Shear Stresses. Annu. Rev. Fluid Mech.* **1**, 111–146, 1969):

$$\tau = \frac{\epsilon}{\sigma}$$

Considering the electrical conductivities experimentally measured from the inks and the dielectric constant of water, the relaxation time of the ink was about 3×10^{-7} s. (The electrical permittivity is the product of the electrical permittivity of vacuum, 8.854×10^{-12} F/m, times the

relative permittivity (dielectric constant) of the ink -here taken to be approximately that of the solvent). On the other hand, considering an electrical conductivity of PEO on the order of 10^{-9} S/cm (Ahmed, H. T. & Abdullah, O. G. Preparation and composition optimization of PEO: MC polymer blend films to enhance electrical conductivity. *Polymers (Basel)*. **11**, 1–18, 2019) and a dielectric constant on the order of 10 (Kliem, H., Schröder, K., & Bauhofer, W. High dielectric permittivity of polyethylene oxide in humid atmospheres. Proceedings of Conference on Electrical Insulation and Dielectric Phenomena - CEIDP '96, 12-15, 1996) we obtained a PEO relaxation time on the order of 10^{-3} s.

When printing PEO patterns, charge dissipates to the high electrical conductivity silicon substrate through wet PEO which has a higher electrical conductivity than dry PEO, thus the effective relaxation times were lower than 10^{-3} s in the range from 10^{-7} s to 10^{-3} s

We improved the discussion originally included in the manuscript by including the experimental values of electrical conductivity measured for our inks (page 12). We also included details of the calculation of the relaxation times, as Supplementary Note 3. The electrical conductivity of the inks was included in the supplementary information (Supplementary Table 3).

The term of off-plane is confusing. Does the off-plane refer to the Z-axis?

Indeed, the term "off-plane speed" denoted the speed of printing in the Z direction, outside the printing plate. The printing speed in the XY plane is limited by the speed of the jet in the Z direction, but it strongly differs from the printing speed in the Z axis, i.e. the vertical speed of advance of the printed object as layers accumulate, which depends on the printing frequency. More specifically, the printing speed in the vertical direction is the maximum printing frequency times the fiber thickness, and in our experiments reached 2000 layers/second x 0.2 μm = 0.4 mm/s. This speed would be much higher when considering thicker fibers, e.g. a 2 μm fiber oscillated at 2000 Hz would result in a 4 mm/s vertical printing speed. We clarified this point in the manuscript, in the abstract and in page 11.

This work presents an interesting method to quickly generate fine structures, and it definitely has some potential applications, for example, tissue engineering, electrical interconnects, etc. But the reviewer thinks that it is too strong a claim as a 3D printing method, especially when directly compared with conventional 3D printing.

We sincerely thank the reviewer for carefully reading our manuscript and raising his/her concerns, which undoubtedly have helped us to improve the manuscript. The present work demonstrates a proof-of-concept of a novel technology which is at an early stage of development. Like other conventional 3D printing technologies, our technology can print arbitrary 3D designs as the assembly of layers. In our technology, a 3D model is "sliced" into

layers of a height equal to the fiber height. The difference between the technologies lies simply in the form of the ink as it is transported to the substrate, and on the mechanism of transport. While the specific prototype printer demonstrated in the present work still has some limitations, the methodology on which it is based can be used to print predefined patterns. Current limitations to be overcome in our specific printer are: i) to improve the software to be able to compute the jet deflection voltages for printing 3D objects where each layer can be defined from slicing a predefined 3D model; ii) to incorporate a way to quickly interrupt/restart material deposition at will, such as a gutter. None of these limitations is unsolvable. In fact, some demonstrated technology already exists to overcome these limitations. For example, slicing software is a standard feature of any 3D printing technology where layers are printed sequentially. On the other side, gutters have a long history in inkjet printing, where they are used to collect rejected drops in rapid printing of trains of electrically charged droplets. We foresee that with a certain optimization, both existing solutions can be implemented to enhance the capabilities of our 3D printing method. These limitations are stated in pages 5 and 13.

Reviewer #2 (Remarks to the Author):

This manuscript reports on the use of electrohydrodynamic jetting to print 2D and 3D features. The key contribution of this work is the use steering electrodes (i.e. electrodes positioned around the emitted jet) to deflect the jet trajectory at high speed. While past work has used EHD jets to print materials - these previous efforts have generally relied on the use of mechanical stages to create 2D/3D features (which are inherently slow). This work reports on a new capability that combines jet deflection (and the use of stages) to increase printing speed. The authors show that this technique can also exert control over the nanostructure of the printed features by controlling the evaporation rate of the solvent in the jet.

I believe this is good quality work and a novel contribution. The manuscript is generally well-written. The following items should be addressed before consideration for publication.

1. One of the unique features of this technology is the "self aligning" aspect of one layer to the next when printing 3D structures (described on Page 11). While this certainly exists and is reported by others - I am confused by the explanation given in the manuscript. In particular, the use of the term "image charge". From electrostatics, I believe image charge refers to a "fictitious" charge used to solve for a field (for example, the classic solution to the point-plane problem.) In the manuscript - image charge is referring to a charge of opposite sign (to the applied potential, potential of fiber) that collects at the top of the printed features. But why should this happen? The printed fibers should initially have the same potential as the "newly arriving" fiber. This charge on the printed fibers should rapidly dissipate - but it is then replaced with charge of the opposite polarity? Perhaps it is it better to assume that there is electrical continuity from the top of the printed features (to ground)? And if this is the case - would the fibers need to behave as conductors? But if they are polymers - do they not act more like (leaky?) dielectrics? Perhaps all of this can be clarified - as I think it is an important aspect of this work.

We fully agree with the reviewer that the microscopic concept of image charges was not clear and could lead to confusion. We had used the term "image charge" loosely as meaning the charge which is ultimately induced on the fiber as its electrical potential slowly relaxes to that of the substrate. However, as the reviewer points out, this is confusing. To avoid confusion, we now have removed the term "image charge" from the text and, instead, provided the mechanism for the charge reversal (ohmic conduction), along with the explanation that the electrical potential on the printed fiber progressively decreases during the process of charge reversal. Note that the absolute magnitudes of electrical charge in the initial and final states of this process are not the same, nor are their spatial distributions. As noted above, the new text, replacing the old text on page 12, now reads:

After ensuring a fast-enough charge dissipation, the charged nature of the jets was actually highly convenient not only to achieve high printing speeds and accelerations, but also to easily and precisely manufacture 3D objects by self-assembly of the new arriving jet on top of a previously printed structure. Upon reaching the substrate, the electric charge carried by the jet is gradually dissipated by conduction to the substrate and is replaced by charge of opposite polarity, as ohmic conduction through the printed fiber lowers its electric potential (towards 0 V, the potential of the earth-grounded substrate). This opposite-polarity charge accumulates at the top-most surface of the printed object, locally enhancing the electric field. Therefore, the newly arriving jet is electrostatically attracted to this charge, tending to self-assemble with high precision on top of the previously deposited layer. This attraction was fundamental to accurately accumulate layers, overcoming limitations associated to the movement-induced vibrations of mechanical stages²⁰. This is the same mechanism termed "electrostatic autofocusing" by Galliker et al.²⁷ underlying the direct-printing of high-aspect-ratio nanostructures using electrically-charged colloidal nanodroplets.

In addition, we have removed the term previously appearing also in line 362. The new text reads:

Though electrostatic deflection of electrified jets, 3D structures of increasing complexity, including cross overs and bridges, were printed by precise electrostatically-driven layer-by-layer self-assembly at frequencies as high as 2000 layers per second.

2. Following from point 1 above - is it not required that the target substrate be conductive (or at least a semiconductor) in order to print onto it? The initial charge on the printed fiber must be dissipated. This would imply that printing on insulating regions would not be possible (it is suggested it would be possible on page 12).

The substrate needs to be able to dissipate charge sufficiently fast, at a time scale ideally shorter than the time it takes to print a single layer, which we can define as the characteristic printing timescale. Therefore, the required relaxation time of the substrate (and of the printed matter) depends on the layer-by-layer printing frequency, the inverse of the characteristic printing timescale. To dissipate charge a metal or degenerated semiconductor will usually be necessary. When printing at a low frequency, charge can be dissipated during a longer time, and thus a material with lower conductivity, such as glass, could also be used if it can dissipate the charge in the characteristic printing timescale. Generally, to print at any available layer-by-layer frequency on an insulating substrate, such substrate needs to be covered by at least a thin layer of metal. Alternatively, gas phase ions or water vapor could conceivably be injected locally to the printing spot to dissipate charge, although we have not attempted to develop this approach. We have introduced a discussion on the possible substrates on page 4. Also, we have included a new SEM image showing lines printed on paper sputter-coated with a thin layer of silver (Supplementary Figure 4).

3. For this printing methodology to work, the contact point between the jet and substrate and the jet speed must be equal. Surely achieving this balance must be difficult. The jet speed and deflection must (independently) be related to a number of factors (conductivity of solution, applied potential, etc.). But the dependence of jet speed and deflection on these factors is only briefly addressed (this should be expanded on). In addition - can a statement be made on how difficult it is to equate the jet speed and contact point speed for printing.

The reviewer is correct: To print patterns with high precision, the contact point speed and the jet speed must be similar. Significantly higher contact point speeds result in patterns with rounded corners, while significantly lower contact point speeds result in fiber buckling. As noted by the reviewers, the jet speed and contact point speeds depend on several factors, but for a specific ink and given operating conditions, the jet speed is approximately constant in time. Therefore, in a hypothetical commercialization scenario, each ink would be operated at its optimum operation conditions and printing speed. In the laboratory, we change inks relatively frequently for research purposes. When testing a new ink, the main challenge is not the matching of speeds, but to ensure jetting stability and constant jet speed over a long time, which requires optimization of ink composition, ink flow rate, and of solvent evaporation rate. To match both speeds is a simple task when using a macroscopic calibration pattern that allows to experimentally determine the jet speed. Once the jet speed is known, it is trivial to find the appropriate jet-deflection parameters to ensure that deflection speed matches the jet speed. The dependence of jet deflection speed on deflection parameters, such as signal amplitude and frequency, is demonstrated in Fig. 2c. More precisely, for a given pattern, first the signal amplitude is selected providing the desired pattern size. Then, the software computes the length of fiber necessary to print one desired pattern. The experimentally obtained value of jet speed is then divided by the fiber length necessary to print one pattern, giving the frequency. The current version of our software allows selecting the printing speed by presetting pattern geometry, amplitude and frequency. However, future version can implement automatic computation of amplitude and frequency based on the desired pattern. This is a trivial computation which can be simply realized by mapping the jet deflection angle dependence on signal amplitude and frequency, as demonstrated in Fig. 2c.

Following the reviewer suggestion, we expanded the discussion of the dependence of the jet speed on the different factors in page 9 and included a statement on how the two speeds are balanced and on the difficulty of doing so in page 8.

4. A minor point: what would be the largest feature that can reliably be printed? I understand this can likely be calculated from the separation distance and deflection angle - but an explicit statement may be appropriate.

As noted by the reviewer, the largest feature that can be printed by just jet deflection depends mainly on the distance between the nozzle and the substrate (nozzle elevation). With this distance set at 5 mm, we obtained a maximum printing area of 2 x 2 mm². This area can be increased by placing the nozzle at a larger distance from the substrate. The electrodes configuration (1 or 2 electrodes per deflection axis), their position and design, and maximum deflection potentials are also important parameters which affect deflection angle, and ultimately the printing area at a fixed nozzle elevation. As noted in the manuscript, we prefer using deflection angles smaller than 15 degrees because the resulting deflection on the substrate is linear with the amplitude of the applied jet deflecting signal (Fig. 2a-b). To print patterns with substantially larger angles, the software must be programmed to account for the nonlinear dependence of deflection with amplitude. To clarify these points, an explicit statement of the largest printed feature using jet deflection was introduced in page 6.

5. While this technology has the advantage of speed - one potential disadvantage over other printing technologies is the apparent inability to print sacrificial materials (for example, to print interior features, etc.) Certainly this is an important capability of 3D printing. Can the authors address this point?

The developed technology can print sacrificial materials such as polymers that work as templates to produce other structures, which can afterwards be dissolved away (as in Kotz, F. *et al.* Fabrication of arbitrary three-dimensional suspended hollow microstructures in transparent fused silica glass. *Nat. Commun.* **10**, 1–11, 2019). The technology can also print different materials, so it could combine the printing of a sacrificial material with another material to define an object. To implement this multi-material printing would involve a mechanism to align with sub-micrometer resolution the substrate so that a second material is printed in a predefined position with respect to the material firstly printed. This multi-material printing capability needs to be developed in future prototypes.

REVIEWERS' COMMENTS:

Reviewer #1 (Remarks to the Author):

The authors have addressed all the questions in this revision. The reviewer has no further questions or comments.

Reviewer #2 (Remarks to the Author):

All of my comments have been suitably addressed. I recommend this manuscript for publication.